# Biopolymer Composites as an Alternative to Materials for the Production of Ecological Packaging

**DOI:** 10.3390/polym13040592

**Published:** 2021-02-16

**Authors:** Miroslawa Prochon, Oleksandra Dzeikala

**Affiliations:** Institute of Polymer and Dye Technology, Faculty of Chemistry, Lodz University of Technology, Stefanowskiego 12/16, 90-924 Lodz, Poland; dzeikala.sandra@gmail.com

**Keywords:** gelatin, PVA, composite, biodegradation, packaging

## Abstract

The problem of plastic waste has long been a concern for governments and society. However, huge amounts of plastic are still being released into the oceans and the environment. One possible solution is to replace plastics with materials that are more both biodecomposable and biodegradable. The most environmentally friendly materials are made of natural ingredients found in nature, although not all of them can be called biodegradable. In this study, we set out to create a new composite with functional properties that could replace commonly used disposable packaging. To ensure the competitiveness of our solution, we used inexpensive and readily available components, such as gelatin *G* HOOCCH_2_CH_2_C(R_1_)NHCOCH_2_NH_2_ (where R_1_ is a continuation of the peptide chain), polyvinyl alcohol *PVA* CH_2_CH(OH), and glycerin *G* HOCH_2_CH(CH_2_OH)O. The ingredients used in the research come from natural sources; however, they are chemically processed. Some of them, such as polyvinyl alcohol, for example, are biodegradable. With the appropriate selection of the components, in the casting process, the intermixed components made it possible to produce materials that were characterized by good physicochemical properties, including thermal stability, optical transmission of UV-Vis light, cross-linking density, and mechanical strength. The most favorable parameters of thermal stability were observed in casein-containing gelatine forms. The best cross-linking density was obtained in the case of gelatin–glycerine systems. Composite containing caseins distinguished by the highest resistance to flammability, increased thermal stability, flexibility, and greater hardness compared to other composites.

## 1. Introduction

Packaging is used for the storage, protection, handling, delivery, and presentation of goods, from raw materials to finished products. In the food industry, packaging helps to protect products from chemical, mechanical, and microbiological damage, as well as ensuring the freshness of the product and preserving its nutritional value [1,2,3]. Most of the packaging used today is synthetic and based on fossil fuel. However, there is widespread concern about the depletion of non-renewable raw materials for the production of plastics. Moreover, there is growing awareness of the problem of plastic pollution, and its adverse effects on health and the environment [4,5,6]. On average, more than 75,000 plastic microparticles are inhaled or ingested by Americans each year. Plastic may enter the human body with food, such as fish, or in bottled water [7,8,9,10]. A comparable number of particles and fibers enter the body with air [11,12]. The majority of microplastics (54%) are polypropylene, from which bottle caps are made, and 4% of these showed the presence of industrial lubricants [12,13,14,15]. If current trends continue, some experts believe polymer particles will soon be found in all food products. One solution is to produce less plastic, which could bereplaced with more environmentally friendly polymers [12].

Materials entirely composed of biomaterials can be grouped into three main categories, according to their origins and production methods. Polymers extracted/isolated directly from biomass are the most common on the market. Polymers in this category are obtained from plants, as well as marine and domestic animals. Examples include polysaccharides such as cellulose, chitin, starch, and polypeptides, such as whey protein, casein, collagen, and soy protein. Other examples are regenerated cellulose film (cellophane paper) and cellulose acetate [16]. It is also possible to obtain a wide range of bio-polyesters via classical chemical synthesis and from monomers. The most famous biopolymer from this group is polylactic acid (polylactide, PLA). In terms of mechanical and strength properties, PLA is similar to polystyrene; however, when modified, it has properties similar to polypropylene and polyethylene. PLA has the ability to stretch crystallization, to temperature crystallization, it can be impact-modified, copolymerized, and processed on most processing equipment. It can be used to obtain transparent films or injection molded into blow molds, such as PET polyester (polyethylene terephthalate). It has excellent organoleptic properties and is ideal for contact with food. The raw material for lactic acid is produced by the fermenting of glucose or starch. Corn, wheat, whey, or molasses can be used as sources of carbohydrates [17,18]. The third group is polymers obtained directly from natural or genetically modified organisms [19]. This group includes polyhydroxyalkanoates (PHA) and bacterial cellulose. PHAs are polyesters that are part of the living structure of the organism. They are hydrophobic and insoluble in water. Their properties are most closely related to those of their monomer building blocks. A wide variety of biopolymers can be synthesized by microbial fermentation [20]. Biodegradable packaging is still largely at the research stage. Theoretically, all currently used packaging materials could be replaced with renewable monomers. However, there remains the question of economic viability [21]. In the case of polymer packaging, the term biodegradation used to be associated with mixing polyolefins with polymers of natural origin, such as starch, cellulose, etc. [22]. This was important for the growth of microorganisms on the surface of such a material or was associated with a loss of mechanical properties (e.g., tensile strength). Currently, efforts are being made to ensure that there is not only the biodegradation and dispersion of harmful synthetic components in the natural environment, but above all that the polymer material does not contain synthetic components and that their complete deterioration in the environment takes place to decomposition products such as CO_2_, water, or compost constituting medium for the growth of other plants [23,24]. Full commercial production and the use of biodegradable packaging is still in most cases a distant goal. There remain a number of challenges to be overcome, such as achieving cost effectiveness, satisfying food safety requirements, ensuring quality standards, improving the water vapor barrier, and many others. Food packaging also has to meet strict regulatory and safety requirements and quality standards. It must also be functional and often aesthetically attractive. However, the potential of biomaterials is already clear. For example, biomaterials have been made containing pregelatinized starch (40–70%), with the addition of a hydrophilic copolymer such as ethylene acrylic acid, polyvinyl alcohol, or vinyl acetate for thickening. Starch takes 40 days to completely degrade, and the entire film takes 2–3 years [16]. In comparison, materials containing synthetic compounds, so-called plastics, decompose over hundreds of years in the natural environment. The production of modern materials is also associated with the use of simple processing methods. In addition to methods such as extrusion, calendering, coating, dipping, sintering, and injection, casting methods, for example solvent casting, are also used. Solvent casting can be by gravity or rotation. The first takes place with the use of casting machines, while the second, rotary casting, runs along the walls of the mold dedicated to the appropriate shape of the finished product. The solvent casting method consists of dissolving the polymer in, for example, an organic, polar solvent and adding particles of specific dimensions to the solution. Then, the mixture is formed into the final geometric form, for example on glass or Teflon plates, preventing the adhesion of the material itself to the substrate. When the solvent evaporates, a composite material structure is formed containing the particles together with the polymer [25,26,27]. The aim of the research was to produce biopolymer composites using the casting method, with increased pro-environmental properties from natural resources. The produced biopolymer coatings do not contain synthetic polyolefins, such as polypropylene (PP) or polyethylene (PE); their main matrix is made of partially biodecomposition and partially biodegradable materials such as gelatin, starch, casein, and poly (vinyl alcohol) (PVA). Various additives were applied to the base composition to improve its properties. The obtained polymer films were found to have good physico-chemical properties (higher tensile strength, higher hardness, improved thermal stability, etc.), and they could be used as packaging material. 

## 2. Materials and Methods

### 2.1. Research Facilities

The polymer matrix was gelatin (pro analysis (p. a.), Mw 30 kDa, pH 4–8, Bloom 200, FoodCare Sp.z. O.o., Zabierzów, Poland) and glycerin (p. a., (Mw 92 kDa; pH 5.5–8; relative density 1.26 g/cm^3^, flash-point 160 °C, CHEMPUR, Piekary Śląskie, Poland). In addition to gelatin, substances such as casein were also used (p. a., Mw 75 kDa, ≥95% alkali soluble, bulk density 450 kg/m³, water solubility 20.1 g/L, Sigma-Aldrich Sp. Z oo, Poznań, Poland), potato starch (p. a. Mw 162 kDa; pH 6–8; water solubility 50 g/dm^3^, bulk density 280 kg/m^3^ Kupiec sp.z.o.o., Krzymów, Poland), poly (vinyl alcohol) (PVA) (pur., Mw 20–30 kDa, pH 4.5–6.5; relative density 1.3 g/cm^3^, flash-point >180 °C CHEMPUR, Piekary Śląskie, Poland) and red phosphorus (Organic Chemistry, Straconki, Poland). The following natural dyes were used to color the composites: brilliant blue FCF E133; pur. quinoline yellow E104; cochineal red E124 (purum pur. 1-(1-naphthylzo)-2-hydroxy-naphthalene-4′, 6.8 trisodium trisulfonate (pur. Molecu SYNCHRO, Lodz, Poland). Coloring was an intended goal due to the future use of the produced films as packaging or prefabricated items for packaging. Manufacturers often strive to maintain the appropriate aesthetics of their products. However, the use of dyes was important for the quick organoleptic differentiation of the composition of raw materials of individual materials.

### 2.2. Preparation of the Composites

A reaction set equipped with a three-necked flask, a stirrer, a thermometer, and a reflux condenser was used to prepare the composites (Figure 1). A thermal chamber (Binder GmbH, Tuttlingen, Germany) and a hydraulic press (Skamet 54436, SKAMET, Skarzysko-Kamienna, Poland) were used for drying and shaping. The following materials were used to characterize the composites: a Zwick 1435 testing machine (Zwick/Roell, Radeberg, Germany); the Nicolet 6700 spectrophotometer (Thermo Scientific, Waltham, MA, USA); a Shore hardness tester type A with a pressure force of 12.5 N, indenter 35°Sh, spring force 806.50cN (Zwick/Roell, Herefordshire, Great Britain); an OCA 15EC goniometer (Dataphysics, Filderstadt, Germany); a DSC1 differential scanning calorimeter (Mettler Toledo, Netzsch, Switzerland); and a UV-VIS CM-3600d spectrophotometer (Konica Minolta, London, UK.). Scanning electron microscopy (SEM) Zeiss Ultra Plus Bruker, Massachusetts, USA) was used to identify structural changes in the composites. An HPP 108 climatic chamber (Memmert GmbH, Schwabach, German) was used to determine the biodegradation. The compositions of the reagents used are given in Table 1 and Table 2. 

### 2.3. Selection of Matrix Composition

In the first stage of the research, we prepared polymer matrixes with the compositions given in Table 1. The solutions were prepared by mixing the raw materials in appropriate weight ratios.

Most of the chemical reactions were carried out in a standard reaction set (Figure 1), including a three-necked flask and an Allihn reflux condenser. The reaction mixture was heated to 70 °C in an oil bath on a temperature-controlled magnetic stirrer.

Most of the chemical reactions were carried out in a standard reaction set (see Appendix A), including a three-necked flask and an Allihn reflux condenser. The reaction mixture was heated to 70 °C in an oil bath on a temperature-controlled magnetic stirrer.

### 2.4. Preparation of Compositions Based on Gelatin

The preparation process is extremely important for obtaining good quality films. Figure 2 presents compositions with different mass fractions of gelatin enriched with polyvinyl alcohol and starch. The appropriate amounts of glycerin, gelatin, and distilled water were placed in a three-necked flask equipped with a reflux condenser, a stirrer, and a thermometer. The reaction solution was concentrated and dried to form a precomposite. Then, the precomposite was formed into a foil shape, from which samples of various shapes were cut for further research.

The systems were stabilized with glycerin, which was used as a hygroscopic and humectant substance. Due to the presence of three hydroxyl groups, glycerin is very miscible in water. After the synthesis reaction—combining the components, carried out in a chemical reactor for 2 h—the solution was concentrated, stabilized in a thermoformed chamber, and shaped at specific temperature and pressure parameters.

### 2.5. Introduction of the Filler 

In the second stage of the research, various additives were applied to the selected gelatin substrate in order to improve the film-forming properties of the polymer matrix. Casein and red phosphorus are used as fillers due to their flame-retardant properties. PVA and potato starch were used as filling additives, and glycerin was used as a plasticizer to provide the desired flexibility of the composites. The addition of 5 parts by weight of casein, starch, PVA, or phosphorus, respectively, is sufficient to ensure the desired final properties of the produced gelatin–glycerine composite films, for example, such as mechanical strength or thermal stability. The ingredients of the compositions are given in Table 2.

The appropriate filler (Table 2) and dye (quinoline yellow or cochineal red) were added to a reaction mixture containing gelatin and stabilizer. The process was carried out for 1 h. The contents of the flask were concentrated on a rotary evaporator (Rotavapor® R-300, Buchi, Warszawa, Poland) and thermostabilized in a thermal chamber (see Section 2.2). The composites formed under specified pressure and temperature (Figure 2).

### 2.6. Research Techniques 

The composites containing natural fillers were analyzed to determine their tensile strength (TSb, MPa), relative elongation at break (Eb, %), and hardness. Measurements were made in accordance with PN ISO 37: 1998 (D1235 ASTM) using paddle-shaped samples. The hardness of the composites (ISO 7619-1 standard) was determined using a digital Shore A hardness tester, scale A. FTIR spectroscopy was used to determine the characteristic vibrations for groups occurring in the biopolymer compounds. FTIR was used to determine the changes that occurred in the tested composites in comparison to the reference sample, following the addition of fillers to the gelatin matrix. Measurements were carried out in the radiation wavelength range of 4000–400 cm^−1^ with a resolution of 0.25 cm^−1^.

The structures were observed using a Philips XL30 Environmental Scanning Electron Microscope (ESEM) at 10 kV. Differential scanning calorimetry (DSC) was used to determine the changes in heat capacity (∆Cp) in J/g∙K of the selected composites and to estimate the glass transition temperature (Tg), using a computer program Mettler Toledo TGA/DSC (thermogravimetry/differential scanning calorimetry) analyzer (Mettlet Toledo, Greifensee, Switzerland) calibrated using the standard pattern (indium, zinc). The standard deviations of the results were in the range of Tg ± 20 °C. Measurements using DSC are visualized as a thermogram showing the DSC curve, showing the dependence on temperature of the measured differences in heat flux. A constant heating rate of 100 °C/min was used for the measurements.

The Owens–Wendt–Rabel–Kealble (OWRK) method was used to determine the Surface Free Energy (SFE) of the tested compositions. This method involves measuring the contact angle of the tested material using two measuring liquids, one of which is polar and other of which is non-polar. The polar liquid was distilled water. The dispersion liquid was diiodomethane. Based on Equation (1), the polar component $\gamma_{S}^{p}$ describing the sum of the forces of hydrogen, acid–base, and inductive interactions was calculated, together with the dispersion component $\gamma_{S}^{p}$ defining the London dispersion force:(1)$$\frac{\gamma \left(1+cos\theta \right)}{2}=\sqrt{\gamma_{S}^{d}\gamma_{L}^{d}}-\sqrt{\gamma_{S}^{p}\gamma_{L}^{p}}2$$
where q is the contact angle on the tested surface of the measuring liquids [0], γL is the surface free energy of the measuring liquid [mJ/m^2^], γS is the surface free energy of a solid body [mJ/m^2^], and γSL is the surface free energy at the solid–liquid interface [mJ/m^2^].

The values for $\gamma S_{p}$ and $\gamma S_{d}$ are substituted into Equation (2), and the SFE of the tested compositions is calculated:(2)$$\gamma S=\gamma S_{p}+\gamma S_{d}.$$

The measurement was made in accordance with the PN–ISO 1817: 2011/Ap1: 2002 standard. Samples weighing from 30 to 40 mg were cut from each composition and then placed at room temperature for 48 hours. The weights of the swollen and dried samples were used to calculate the following parameters: Equilibrium swelling by weight Qw:(3)$$Q_{w}=\frac{m_{sp}-m_{s}}{m_{s}}$$

*m_sp_*—weight of the swollen sample [mg]

*m_s_*—mass of the sample dried after swelling [mg]

Degree of cross-linking *α_c_*:(4)$$\alpha_{c}=\frac{1}{Q_{w}}\;.$$

To estimate the effect of thermo-oxidative aging on the composites, measurements were made in accordance with the PN-821C-04216 standard. Paddle-shaped samples were placed in a recirculating air binder chamber for a period of seven days at a temperature of 70 °C. Soil tests were carried out in a Memmert climatic chamber in accordance with the PN-EN-ISO 846 standard to determine the susceptibility of the composites to biodegradation. Paddle-shaped samples were placed in soil for 14 days at a temperature of 30 °C with 80% air humidity. Color change as a result of external factors was determined using the spectrophotometric method in accordance with the PN-EN-ISO 105-J01 standard. The CIE Lab color space (UV-VIS spectrophotometer) was used.

## 3. Results and Discussion

### 3.1. Characterization of Base Composites

#### 3.1.1. Shore Hardness Testing

The macrohardness measurements presented in Table 3 and Figure 3 show that as the gelatin content in the composite increased, the hardness parameter of the tested polymer rose. In the case of systems based on PVA, the addition of starch caused a decrease in hardness of 50%.

Better mechanical properties were shown by gelatin-based composites compared to PVA-based composites. From the data presented in Table 3, it can be concluded that the hardness of the composites increases with the degree of gelatin cross-linking in the sample. Increased hardness and density occurs due to a stronger or denser connection between the matrix and the additives and due to the reinforcement that results from the increases in holding time and temperature [28]. An increase in the concentration of gelatin results in biopolymer films with greater density and stability, as reflected in the FTIR analysis.

#### 3.1.2. Breaking Strength

We first tested the static mechanical properties of the base composites, using a testing machine (Zwick 1435, Zwick/Roell, Radeberg, Germany). The results are presented in Table 4, together with the standard deviation.

The systems with the highest concentration of gelatin were the base matrix (G.G.75 composite), which also had the highest mechanical strength. The PVA composites showed the lowest breaking strength (Figure 4a) and the lowest elongation during deformation (Figure 4b). The addition of starch to the system (G.PVA.S) did not cause any major changes in the relative elongation at break.

An increase in elongation was observed in the polymer with higher content of glycerin and water compared to PVA-based composites. This was also indicated by the stress modulus at 100% SE100 elongation. However, the optimal content of gelatin in the polymer systems was 75 PBW. The mechanical strength of polymers is related to the degree of branching in the polymer chains, as well as to the degree of cross-linking itself. Presumably, the content of the crystalline phase and the degree of cross-linking in the gelatin-based composites was higher than that in the PVA-based composite. This improved the hardness of the gelatin-based composites.

#### 3.1.3. FTIR Spectroscopic Analysis

To confirm the structure of the synthesized samples, FTIR measurements were performed for selected G.G.75 composites and the G.G.PVA.20 composite (Figure 5). A significant change in the intensity of some absorption bands suggests the formation of permanent interactions between macromolecules of gelatin, glycerin, and other components. As shown in the exemplary mechanism of the interaction of gelatin, glycerin, and PVA, leading to the formation of a stable structure of the gelatin matrix (Figure 6). According to the literature [29], the FTIR spectra of gelatin extracted from various animals are similar, whose bands were characteristic for peptide bonds. An amide band occurs in the wavelength range of 3600–3100 cm^−1^, which corresponds mainly to the vibrations stretching the N–H bonds, but also O–H, as well as bands for amide I, amide II, and amide III, with wave numbers at approximately 1638, 1533, and 1235 cm^−1^, respectively. Amide band I is mainly attributed to the vibration that stretches the C = O bond. Amide band II corresponds to the coupled bending vibrations of the N–H bond and the tension of the C–N bond. Amide band III comes from the vibration of the stretching C–N bond and the bending vibration of the N–H bond [29].

On the spectra of G.G.75 (Figure 5), as well as signals characteristic of amide bands (Appendix A), additional bands responsible for asymmetrical and symmetrical vibrations were noted for stretching C–H bonds in the CH_2_ groups in the range of 2914 and 2826 cm^−1^.

In the FTIR spectrum of the PVA-based polymer, characteristic bands appeared in the wavenumber range of 3600–3100 cm^−1^, corresponding to the vibration stretching the O–H bond (Figure 5). The maximum of this band at 3286cm^−1^ may indicate the interaction of hydroxyl groups with hydrogen bonds. Already at the stage of introducing glycerin into the composition with gelatin, it is clearly visible that the interaction is enhanced by the increased number of hydroxyl groups in the system (Figure 6). This leads to the creation of new active donor–acceptor sites in the macromolecule of the gelatin matrix [30]. There is a chance for a nucleophilic attack of the introduced PVA macromolecule, which can both attach to the chain structure of the gelatin itself at the carbonyl carbon atom and interact through hydrogen proton exchange or the formation of less durable hydrogen bonds with fragments of glycerin chains. The confirmation of these hypotheses can be found in subsequent studies showing increased wettability of the surface in relation to the polar liquid and thus reduced values of contact angles for the composite containing PVA, and thus lower system stability and lower degree of cross-linking with the gelatin matrix, which is evident in goniometric surface analysis.

#### 3.1.4. Microscopic Research (SEM)

Exemplary microscopic images of the breakthroughs in the G.G.75 composite were obtained with the use of SEM. Figure 7 shows an SEM image of the morphology of the G.G.75 sample. The sample can be seen to be a homogeneous material, made of particles with spherical shapes, which are clearly visible at both lower and higher magnifications. Due to the production process, gelatin is characterized by the presence of entangled clusters, resulting from the reorganization of the heliacal structures that form the polypeptide chains of scleroproteins.

Among the made biopolymer matrices with different concentrations of gelatin used, the most advantageous structure of the decomposition of the additives used was characterized by a composite containing 75 parts by weight of gelatin to 25 parts by weight of glycerin, which is visible in the SEM photos presented. The matrices containing PVA showed worse mechanical parameters and lack of transparency. Therefore, the gelatin–glycerin matrix medium with a gelatin concentration of 75 PBW was used in further analyses to study the effect of adding fillers, plasticizers, film-forming agents, etc. to the composite.

### 3.2. Characterization of the Composites with Fillers

#### 3.2.1. Shore Hardness Testing and Breaking Strength

Table 5 show the mechanical properties of the composites based on the gelatin–glycerin matrix. 

The combination of gelatin with red phosphorus, casein, and PVA (in this case as a filler) improved the mechanical strength of the composites compared to the base matrix composite (G.G.75). Only the combination of the composite with G.G.S.75 starch resulted in a reduction in mechanical properties compared to the other fillers and to the matrix itself. The surface of starch is less polar as shown by the surface analysis, and the films obtained with its participation are characterized by increased brittleness. The greatest hardness was observed for the G.G.P.75 composite, which may be due to the influence of phosphorus on the intermolecular interactions, resulting from the formation of aggregates or new complexes.

#### 3.2.2. FTIR Analysis

The spectra of the gelatin with the addition of various fillers were similar to those for the gelatin of the polymer matrix (Figure 8). However, slight shifts and changes in the intensity of some bands were noted (see Appendix A). 

There were two additional characteristic absorption bands in the FTIR spectrum of gelatin with the addition of potato starch (Figure 8b). The first, appearing in the wavenumber range 3600–3100 cm^−1^, corresponds to the stretching vibrations of O–H bonds. The maximum of this band at 3275 cm^−1^ proves that the hydroxyl groups linked with hydrogen bonds. The second characteristic band, the saccharide band, occurs in the range of 1200–800 cm^−1^, with maximums located at the wavenumbers 1100, 994, and 852 cm^−1^. These correspond to the vibrations of, respectively, bending C–O–H bonds, bending C–O bonds, and symmetrical stretching vibrations of the C–O–C groups [30,31]. The presented mechanism of probable interactions between starch and gelatin (Figure 9) shows vibrations of the C-O-C glycosidic bond, which interacts with the gelatin macromolecule. Ionized connections are formed, which can be associated with the water present in the starch structure due to its hygroscopic nature. Starch may have water contained in the crystal structure, which is not readily available as opposed to water of hydration. This is evidenced by the broad bands of OH groups in the wavenumber range of 3600–3100 cm^−1^ [30,31,32,33].

As a result of introducing casein into the gelatin–glycerin structure, the system was probably stabilized by a higher number of interactions and the formation of hydrogen bonds, as can be seen in the spectrum shown in Figure 8c in the case of the G.G.C.75 composite. The main peaks for the GGPVA.75 composite observed at 3275 cm^−1^ were attributed to the OH stretching vibrations of the hydroxyl group, as well as at 1627 cm^−1^ derived from the alkene group -C=CH (Figure 8d). The consequences of the mechanisms illustrated in Figure 6 are changes in the intensity of the absorption bands shown in the spectrum (Figure 8d), coming from the groups of valence vibrations CO (1627 cm^−1^) or NH (1537 cm^−1^, 1230 cm^−1^), and also OH groups (3286 and 1029 cm^−1^). In the composites with phosphorus, there was a characteristic band at 1030 cm^−1^, belonging to the thick phosphate (Figure 8e) [32,33]. 

#### 3.2.3. DSC and TGA Analysis

DSC analysis was performed on samples containing fillers with flame-retardant properties, including casein and red phosphorus, to determine their phase transformations, such as melting and glass transition. The DSC thermograms for G.G.C.75 and G.G.P.75 differed slightly (Figure 10). The thermal curves followed a similar pattern, and only the minima for the second composite shifted toward lower temperatures (Table 6).

The addition of the casein filler increased the glass transition temperature of the composition more than red phosphorus did (Table 6). Once the glass transition temperature is reached, the energy of rotation around the bonds in the main chain of the composite is reduced. A relatively large endothermic transformation may be observed [34]. Casein, similar to red phosphorus, contains phosphorus atoms in its structure. They occur in the form of ortho and pyrophosphate residues linked with ester, monoester, or diester polypeptide chains. Due to their structure, both additives are included in the group of flame-retardant compounds for polymeric materials, such as melamine, melamine phosphates, poly (melamine phosphates), borates, isocyanurates, poly (ammonium phosphates), etc. There is probably an increase in cross-linking due to the formation of new connections. The addition of a filler hinders and limits changes in the positions of mers and segments in the macromolecules of the polymer.

However, the heat capacity of the composite before and after the analysis (DSC peak) was proportional to the degree of conversion, at 1.253 J/g∙K in the casein composite and 1.113 J/g∙K in the red phosphorus composite. The specific enthalpy of melting was higher for the composite filled with 5 PBW casein, due to the higher glass transition temperature. The lower enthalpy of fusion for the composite with red phosphorus may be due to the fact that it does not dissolve in the polymer matrix but remains in the form of suspended particles [34], as is visible in the SEM photos (Section 3.2.6).

#### 3.2.4. Analysis of the Contact Angle and SFE (Surface Free Energy)

The deposition mechanism of the surface layer determines the morphology of the layer itself and its electrical properties. Therefore, it is important to analyze the physico-chemical parameters of the substrates, in particular the surface energy. In order to determine the surface energy, the contact angle was first measured. The results are presented in Figure 11, which shows the values of the contact angles on the surfaces of the selected composite materials for water and diiodomethane, as well as photographs of the drops of liquids used.

These results were used to determine the surface energy values, as displayed in Table 7.

The composite containing phosphorus G.G.P.75 was characterized by the highest free surface energy. This is probably related to the formation of a complex of phosphorus salts and to the formation of macromolecular connections as a result of non-covalent interactions. The SFE value was 50% higher than in the case of the G.G.S.75 composite with starch [35,36]. Samples of the composites containing PVA and casein show similar SFE surface energy compared to the reference sample.

#### 3.2.5. Measurement of the Equilibrium Swelling

In order to determine the water barrier for the absorption of productive composites, the equilibrium swelling Qw and the degree of cross-linking αc were determined, and the results are presented in Table 8.

The gelatin composition containing phosphorus and polyvinyl alcohol is characterized by lower equilibrium swelling values than samples with starch or casein. During the measurement, the latter composites showed a greater tendency to lose shape. This is due to a lower degree of cross-linking; solvent (water) molecules penetrate the gelatin matrix more easily, breaking some of the bonds. The PVA and phosphorus composites were characterized by a higher degree of cross-linking and more favorable mechanical properties. Thus, the remaining composites show a much stronger hygroscopic effect and, at the same time, a lower degree of cross-linking (α_c_). The incorporation of starch or casein into the structure of the polymer matrix increases the absorption of moisture as well as the absorption of water vapor from the environment, which is manifested by a lower degree of cross-linking and increased flexibility of the structure. This is confirmed by the FTIR analysis (photo 8), which clearly shows the increase in the intensity of the absorption bands in the wavenumber range 2800–3600cm^−1^.

#### 3.2.6. SEM and EDS Analysis of Composites

Macroscopic photos taken with low-magnification devices (Figure 2) did not reveal the topography of the film fractures. This was visualized using SEM (Figure 12). Scanning electron microscopy was used to characterize the interactions between the chains of the biopolymers and the surface morphology of the gelatin film. The photographs of the G.G.P.75 composite clearly show the filler anchored in the matrix. It did not dissolve, yet it reduced the flammability of the composite. This was also confirmed by FTIR analysis, which revealed changes in the absorption bands in the range of 2800–3600 cm^−1^. The introduction of G.G.C.75 starch or G.G.C.75 casein resulted in the formation of spherical cavities due to the creation of pores on the surface of the grains of these fillers. Composite G.G.PVA.75 was characterized by a significantly different surface topography. It formed a continuous, grooved, longitudinal phase, which could be seen at a lower magnification, while at higher magnifications, it revealed surface homogeneity with few artifacts.

Surface analysis by SEM showed the effect of the fillers on the morphology of the gelatin film. The adhesive material influenced the properties of the polymer, including the mechanical and thermal properties measured by DSC. The addition of the fillers to the polymer led to the formation of a thick-film microstructure, due to hydrophilic–hydrophobic filler aggregates that caused discontinuities in the gelatin film matrix. 

In the second part of the analysis, EDS (Energy Dispersive X-ray Spectroscopy) was used to investigate the microstructure of the gelatin composites (Table 9). The results revealed an increase in the content of nitrogen in the case of the composite containing casein. Thus, casein appears to have reduced the flammability of the composites and increase their thermal stability, confirming our previous analyses. On the other hand, the gelatin films containing phosphorus were characterized by an increased content of phosphorus P in addition to 13.6% nitrogen and trace amounts of sulfur.

#### 3.2.7. Determination of Resistance to Thermo-Oxidative Aging, Biodegradation, and Color Change of Composites

##### Thermo-Oxidative Aging

In the next stage of the research, we subjected the samples to a process of accelerated thermo-oxidative aging. After 7 days in a thermo-oxidation chamber, a reduction in the spatial dimensions of the composites was observed, due to thermal contraction. Color discoloration, reduced flexibility, and brittleness were observed compared to the composites prior to aging. For this reason, they were not submitted to further tests of their strength properties. Measurements of the contact angle were performed for composites containing the fillers. Distilled water was used as the measuring liquid. The results are shown in Figure 13. As can be seen, the contact angles for the composites tested before and after thermo-oxidative aging did not change significantly.

Table 10 presents the results of hardness tests for the obtained composites. The aging process affected the Shore A hardness values. In all cases, the hardness parameter increased after incubation in the chamber, which suggests the formation of new reactive sites, such as macro radicals to which other side chains or substituents may attach to form a secondary spatial network.

##### Determining the Susceptibility of the Composites to Biodegradation

Five samples of the composites with paddle-shaped fillers were placed in containers and covered with a layer of universal soil (pH 6.0–7.0). Measurements were made in accordance with the PN–EN ISO 846 standard. The containers were placed in a climatic chamber at a temperature of 30 °C and humidity of humidity to air (WWP) 80%. The measurement period was 30 days. After 30 days, the samples had completely decomposed. We concluded that the composites were characterized by good biodegradability. 

##### Determination of Color Changes after Aging Processes 

Color changes were measured using a CIE Lab spectrophotometer. Samples after thermo-oxidative aging were analyzed, as well as standard samples before aging. The results are presented in Table 11 and in the reflectometric spectra presented in Figure 14.

The refractometric spectra show the dependence of the reflectance [%] as a function of the wavelength [nm]. The dE*ab parameter in Table 11 compares the difference in color before and after aging. Composition G.G.PVA.75 completely degraded after thermo-oxidative aging. The additives had a positive effect on the color changes of the composites. The smallest color change in the aging process was observed for the sample containing red phosphorus. 

The blue curve is characterized by the smallest change in color after the thermo-oxidative aging process.

## 4. Conclusions

The aim of the research was to produce polymer composites from natural resources. Packaging based on natural biopolymers seems to be a good solution to the problem of plastic waste produced on the basis of synthetic polymers and increasingly deposited in various ecosystems around the world. The proposed gelatin systems with the casting method introduced, with additives of natural origin, made it possible to create a stable film-forming coating. The production of porous gelatin composites was based on the economical solvent casting method. In order to increase the biocompatibility of matrices with additives, no organic solvent was used, which is in line with the proclamation of the slogan "green chemistry" and is associated with the limitation of the use of harmful substances. The produced films resulted in stable film-forming matrices that showed increased thermal stability, especially in the case of using casein and red phosphorus as flame retardants for gelatin composites. Especially casein, as a glycophosphoprotein of cow’s milk, affects the above properties due to the presence of phosphate and sugar residues in its structure. The films also showed good mechanical properties and relatively favorable optical transmission properties in the visible light range. This research could help pave the way to improve gelatin yield and lead to the development of new composite materials for packaging applications. These are undoubtedly significant advantages of the polymer films produced in the presented research, which can be successfully used as a material for the direct forming of packaging or as a substitute for the production of laminated tapes. The results of these studies may lead to the development of a composite that combines the best features of each of its basic components, such as mechanical strength, use density, transparency, dye distribution in the polymer matrix, reduced hydrophilicity, and increased biodegradability in the natural environment. The consequence would be the creation of a unique packaging material with appropriate performance parameters, which is suitable for use in various industries. Polypeptide composites are interesting due to their wide availability from various renewable natural sources. However, the major disadvantage of these biopolymers is their sensitivity to water, as their properties can change drastically with their water content. In order to further use these biopolymers as packaging, for example when storing finished products, water must be removed from the structure of these materials. In order to improve the compatibility of the individual components of the polymer matrix, physical and chemical modifications must also be made.

## Figures and Tables

**Figure 1 polymers-13-00592-f001:**
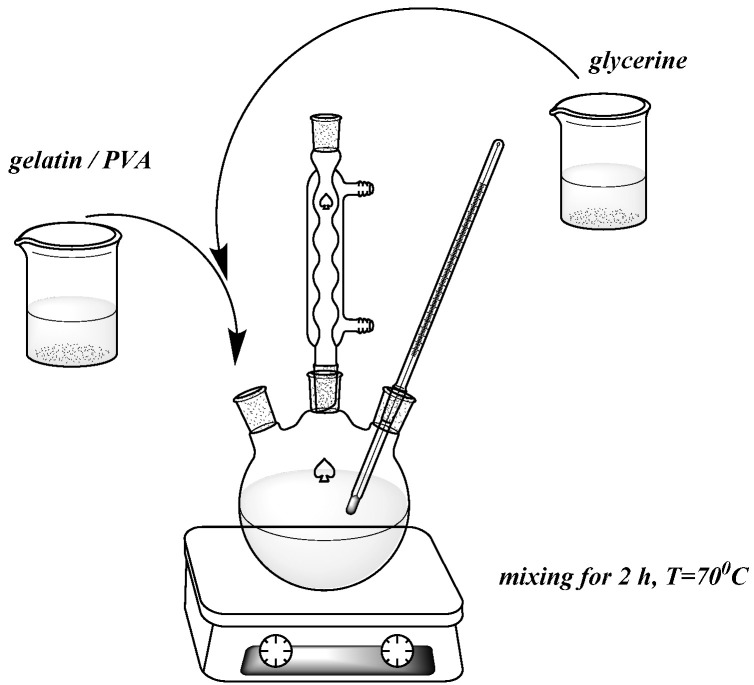
Exemplary scheme of composite synthesis.

**Figure 2 polymers-13-00592-f002:**
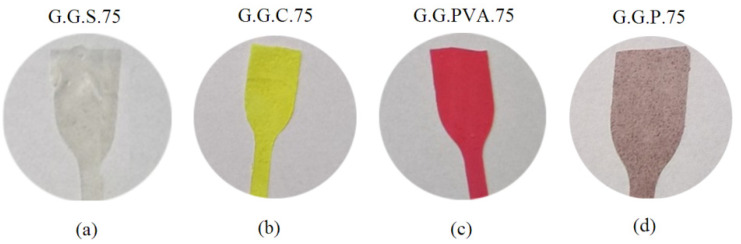
Pictures of compositions with fillers: G.G.S.75 (**a**); G.G.C.75 (**b**); G.G.PVA.75 (**c**); G.G.p.75 (**d**) (private photo resources, CanoScan 4400F).

**Figure 3 polymers-13-00592-f003:**
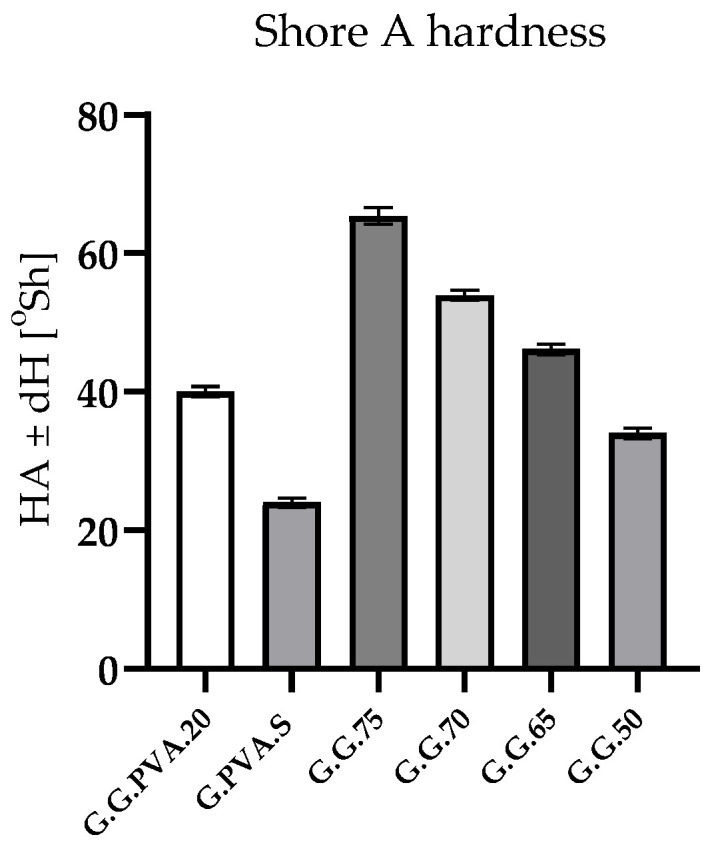
Hardness measurements of gelatin composites.

**Figure 4 polymers-13-00592-f004:**
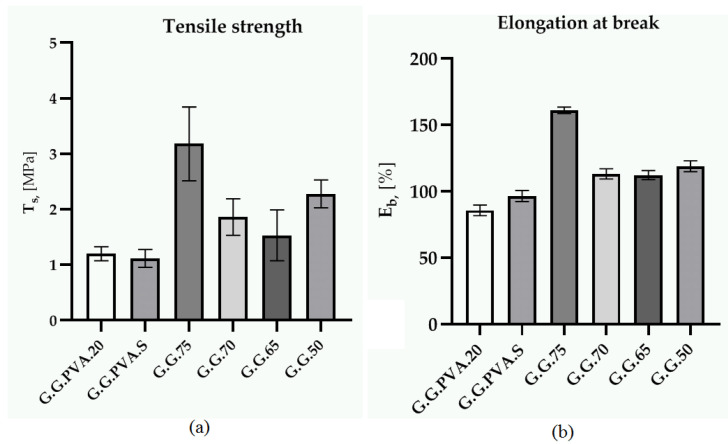
Tensile strength (**a**) and elongation at break (**b**).

**Figure 5 polymers-13-00592-f005:**
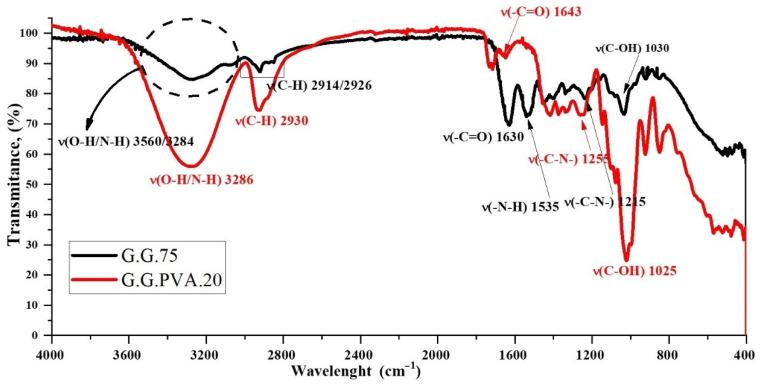
IR spectra of gelatin-based polymer (G.G.75) and poly (vinyl alcohol) (PVA)-based polymer (G.G.PVA.20).

**Figure 6 polymers-13-00592-f006:**
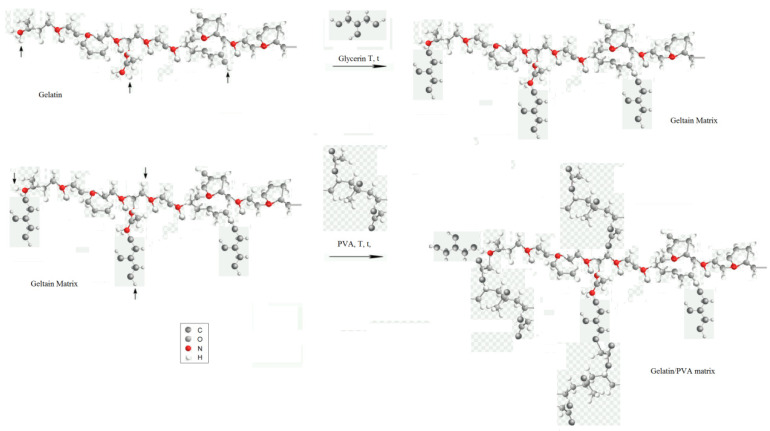
Probable mechanism of interaction of gelatin, PVA, and glycerin macromolecules, leading to the formation of a stable gelatin/PVA matrix.

**Figure 7 polymers-13-00592-f007:**
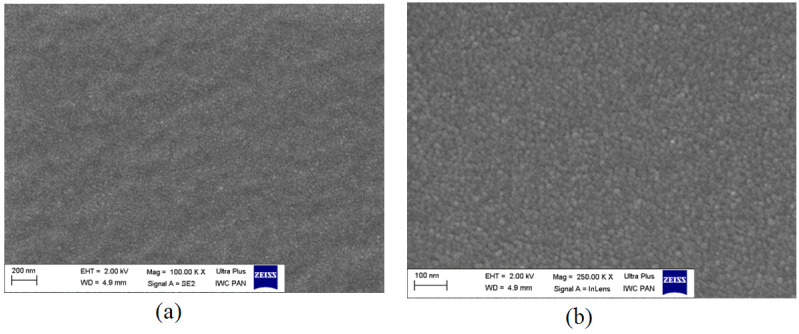
Scanning electron microscope (SEM) image for composite G.G.75 at ×100,000 (**a**) and ×250,000 (**b**) magnification.

**Figure 8 polymers-13-00592-f008:**
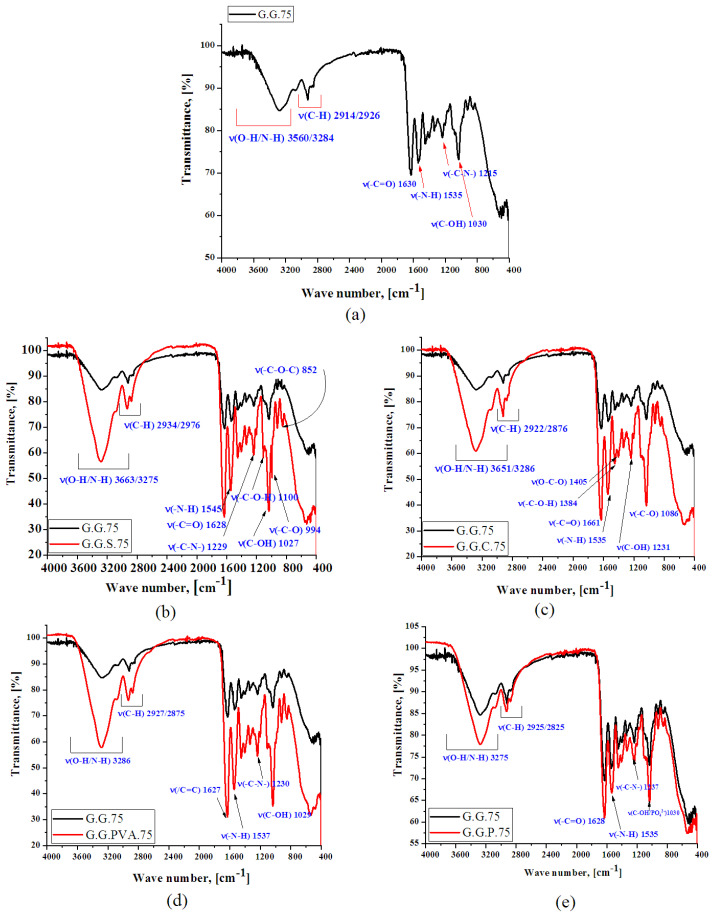
FTIR spectra of gelatin-based composites (G.G.75) (**a**); (G.G.75 and G.G.S.75) (**b**); (G.G.75 and G.G.C.75) (**c**); (G.G.75 and G.G.PVA.75) (**d**); and (G.G.75 and G.G.P.75) (**e**).

**Figure 9 polymers-13-00592-f009:**
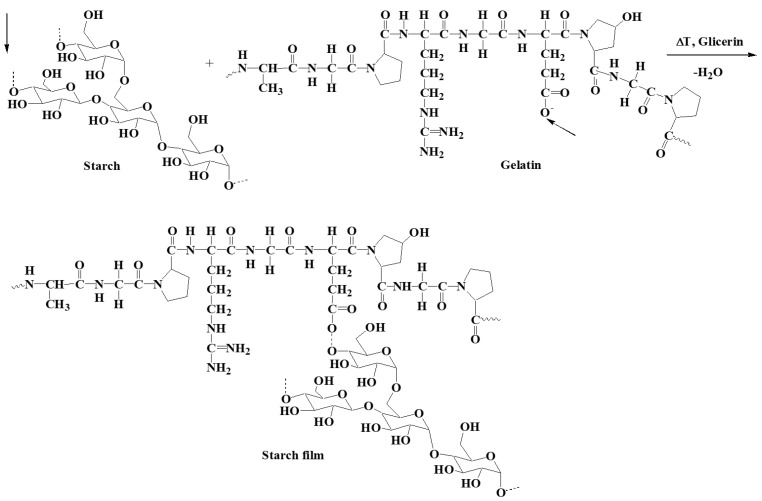
Probable interactions between starch and gelatin macromolecules [30].

**Figure 10 polymers-13-00592-f010:**
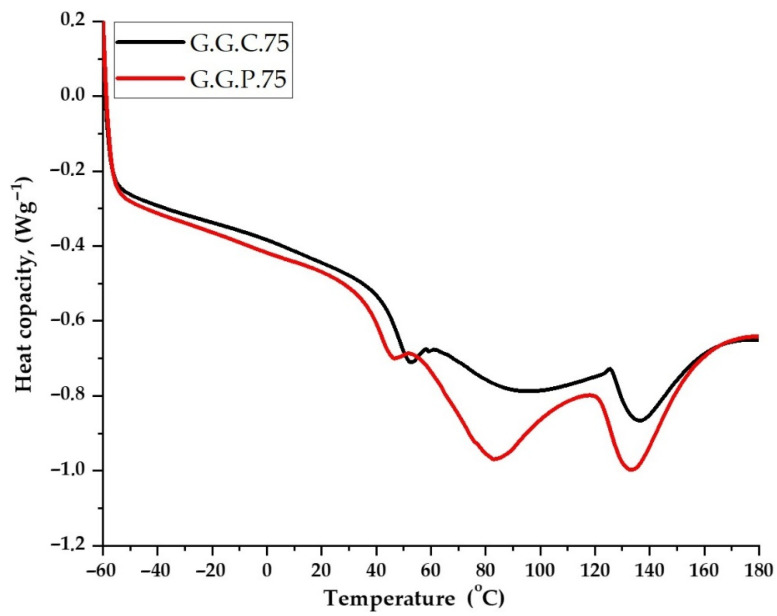
Differential scanning calorimetry (DSC) for composites filled with casein (5 PBW G.G.C.75) and red phosphorus (5 PBW G.G.P.75).

**Figure 11 polymers-13-00592-f011:**
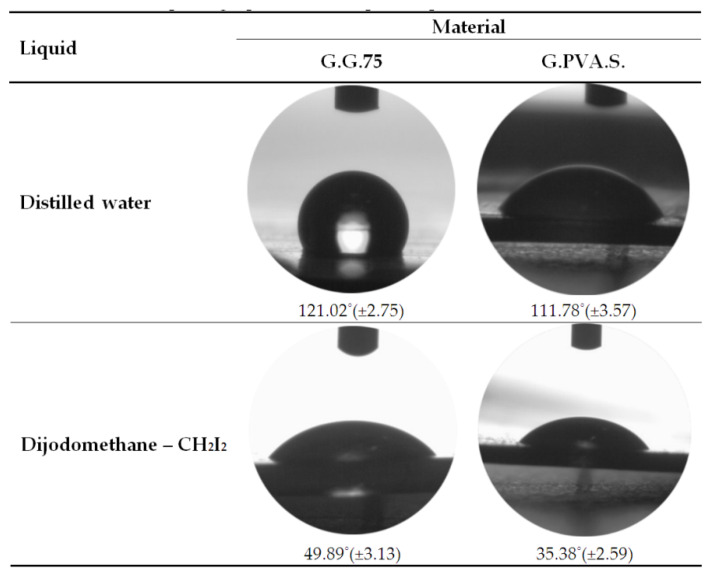
Photographs of water drops and diiodomethane drops on composites: G.G.75, G.PVA.S.

**Figure 12 polymers-13-00592-f012:**
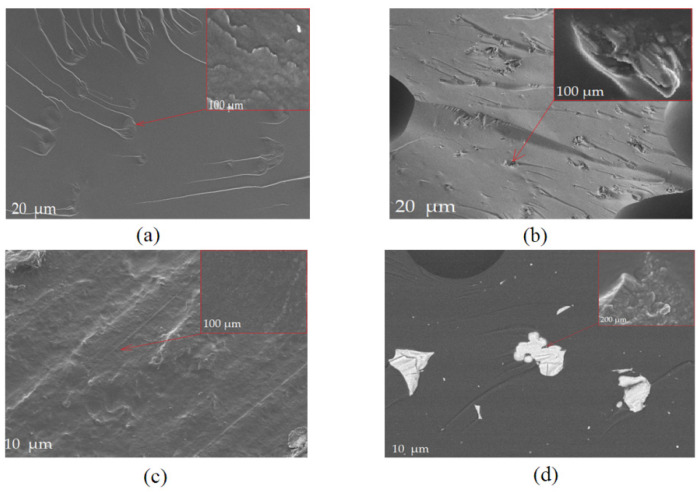
Area SEM pictures of the casein (**a**), starch (**b**), PVA (**c**), and phosphorate gelatin films (**d**), mag. ×1.00, ×25.00 and ×100.00.

**Figure 13 polymers-13-00592-f013:**
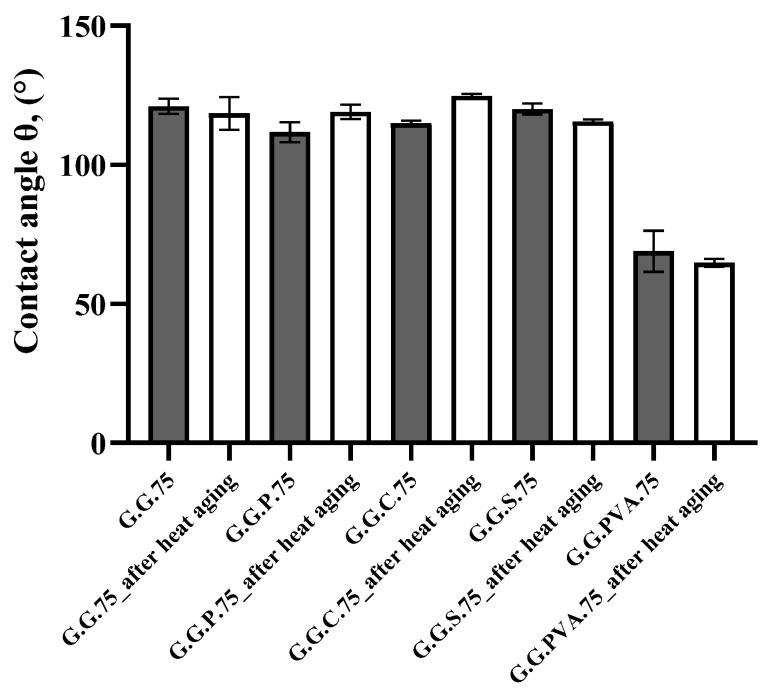
Comparison of the contact angles for samples before and after heat aging.

**Figure 14 polymers-13-00592-f014:**
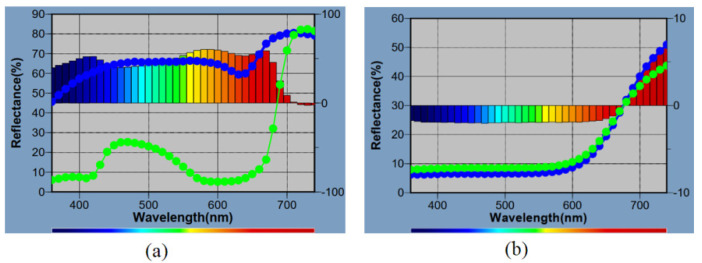
Reflectometric spectrum of G.G.75—reference (**a**) and G.G.P.75 (**b**).

**Table 1 polymers-13-00592-t001:** Compositions of the matrixes.

Composition	Ingredients/PBW				
	Gelatine [PBW]	Glycerine [PBW]	PVA [PBW]	Starch [PBW]	Distilled Water [PBW]
**G.G.PVA.20**	20	10	50	*-*	20
**G.PVA.S**	*-*	20	8	20	60
**G.G.75**	75	25	*-*	*-*	83
**G.G.70**	70	30	*-*	*-*	78
**G.G.65**	65	45	*-*	*-*	72
**G.G.50**	50	10	0	0	125

Legend: G.G.PVA.20—composite with polyvinyl alcohol additive; G.PVA.S—composite containing polyvinyl alcohol and starch, G.G.75; G.G.70; G.G.65; G.G.50—composites with different gelatin concentrations, in PBW (Parts By Weight).

**Table 2 polymers-13-00592-t002:** Composition with fillers expressed in PBW.

Composition	Gelatine G [PBW]	Glycerine G [PBW]	Starch S [PBW]	Casein C [PBW]	PVA [PBW]	Phosphorus P [PBW]
**G.G.S.75**	75	20	5	0	0	0
**G.G.C. 75**	75	20	0	5	0	0
**G.G.PVA.75**	75	20	0	0	5	0
**G.G.P.75**	75	20	0	0	0	5
Other ingredients: Distilled Water—83 PBW						

**Table 3 polymers-13-00592-t003:** Shore A hardness measurements with standard deviation.

Composition	H_A_ ± dH [°Sh]
**G.G.PVA.20**	40.14 ± 0.54
**G.PVA.S**	24.00 ± 2.02
**G.G.75**	65.36 ± 1.13
**G.G.70**	53.92 ± 4.64
**G.G.65**	46.38 ± 2.56
**G.G.50**	34.00 ± 1.61

**Table 4 polymers-13-00592-t004:** Results of mechanical measurements.

Compositions	Parameters			
	TS_b_ [MPa]	E_b_ [%]	SE_100_ [MPa]	a_0_ [mm]
**G.G.PVA.20**	1.20 ± 0.21	84.9 ± 3.7	-	0.46
**G.PVA.S**	1.12 ± 0.36	96.4 ± 4.7	-	0.65
**G.G.75**	3.18 ± 0.75	161.0 ±2.8	2.51 ± 0.61	0.67
**G.G.70**	1.86 ± 0.41	113.1 ± 4.6	2.16 ± 0.43	0.31
**G.G 65**	1.53 ± 0.54	112.2 ± 4.1	1.96 ± 0.53	0.37
**G.G. 50**	2.28 ± 0.27	118.6 ± 3.9	1.72 ± 0.52	0.57

Legend: TS_b_—tensile strength in MPa_,_ E_b_—relative elongation at break in %_,_ SE_100_—stress at 100% elongation in MPa_,_ a_0_—film thickness in mm.

**Table 5 polymers-13-00592-t005:** Mechanical properties of the composites with different types of fillers.

Composites	Parameters (±S.E.M.)				
	H [°Sh]	TSb [MPa]	Eb [%]	SE100 [MPa]	a_0_ [mm]
**G.G.75—reference**	65.36 ± 1.13	3.18 ± 0.75	161.0 ± 2.8	2.51 ± 0.61	0.67
**G.G.S.75**	44.92 ± 1.03	5.40 ± 0.31	15.70 ± 0.4	1.55± 0.80	0.20
**G.G.C. 75**	85.38 ± 1.19	8.08 ± 0.57	139.6 ± 3.2	6.20 ± 0.55	0.21
**G.G.PVA.75**	77.02 ± 1.47	6.71 ± 0.18	138.3 ± 0.5	5.29 ± 0.72	0.14
**G.G.P.75**	92.26 ± 2.63	10.88 ± 0.45	75.40 ± 2.1	7.03± 0.65	0.13

**Table 6 polymers-13-00592-t006:** Thermal properties of samples containing fillers with flame-retardant properties.

Composites	Tg [°C] (ONSET)	Tg [°C] (MILDPOINT)	∆Cp [J/g∙K]
**G.G.C.75**	41.46	48.02	1.253
**G.G.P.75**	35.78	41.54	1.113

Legend: Tg—glass transition temperature, °C; ∆Cp—heat capacity, J/g∙K.

**Table 7 polymers-13-00592-t007:** Comparison of the polar component, dispersion component, and surface energies of samples.

Composites	Polar Component $\gamma_{S}^{p}$ [mJ/m^2^]	Dispersion Component $\gamma_{S}^{d}$ [mJ/m^2^]	Surface Energy *gS* [mJ/m^2^]
**G.G.75—reference**	5.23	43.34	48.57
**G.G.S.75**	1.01	24.19	25.20
**G.G.C.75**	3.02	41.77	44.79
**G.G.PVA.75**	5.05	40.14	46.20
**G.G.P.75**	3.89	49.80	53.69

**Table 8 polymers-13-00592-t008:** The equilibrium swelling Q_w_ and the degree of cross-linking α_c_.

Composition	Q_w_	α_c_
**G.G.S.75**	40.56 ÷ 0.03	5.53 ÷ 0.05
**G.G.C. 75**	35.45 ÷ 0.10	5.82 ÷ 0.04
**G.G.PVA.75**	28.03 ÷ 0.08	8.30 ÷ 0.09
**G.G.P.75**	25.11 ÷ 0.15	9.67 ÷ 0.05

**Table 9 polymers-13-00592-t009:** Composite microstructure analysis of G.G.C.75 and G.G.P.75 percentage distribution of elements in the study of breakthroughs of gelatin composites.

	G.G.C.75		G.G.P.75	
Element	Mass [%]	Atom [%]	Mass [%]	Atom [%]
**Carbon**	48.48	54.67	54.58	61.75
**Oxygen**	34.04	28.82	27.01	22.94
**Phosphorus**	0.49	0.21	4.48	1.97
**Nitrogen**	16.73	16.18	0.32	0.14
**Sulfur**	0.26	0.11	13.62	13.21

**Table 10 polymers-13-00592-t010:** Shore A hardness measurement results with standard deviation.

Composites	Before Aging	After Aging
	H_A_ ± dH [°Sh]	
**G.G.75—reference**	65.36 ± 1.13	73.18 ± 2.15
**G.G.S.75**	44.92 ± 1.03	51.16 ± 1.94
**G.G.C. 75**	85.38 ± 1.19	99.01 ± 2.34
**G.G.PVA.75**	77.02 ± 1.47	-
**G.G.P.75**	92.26 ± 2.63	99.74 ± 1.85

**Table 11 polymers-13-00592-t011:** Color changes of samples subjected to thermo-oxidative aging.

Composition	dE*ab
**G.G.75—Standard**	48.86
**G.G.S.75**	7.44
**G.G.C. 75**	12.59
**G.G.P.75**	4.29

## Data Availability

The data presented in this study are available on request from the corresponding author.

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
