# Peer review of "Biopolymer Composites as an Alternative to Materials for the Production of Ecological Packaging"

_polymers, 2021, doi:10.3390/polym13040592_

Round 1

Reviewer 1 Report

The authors describe the preparation and characterization of composites for packaging. Their work is interesting and could be published after major revision, which include text rewriting and data not presented in this version.

Hereby a non-exhaustive list of comments:

  • In the introduction the definition of biopolymers given seems to imply that they are all biodegradable. The authors should better differentiate between bio-sourced and bio-degradable.
  • the materials and method section is too long and the structure is not clear. I would like a complete rewriting of it, for example section on composite preparation, mechanical charcterization, surface characterization, etc eg: explaining the Owen-Wendt method is out of scope.
  • Section 3 is a results section. I did not see any proper discussion. Composite with gelatin are these properties, while with starch has others. The underlying reason is never even approached, why a specific component has that effect? Rely on previous literature and/or make  aducated hypothesis to provide explanations.
  • 3.1.1 please add a graph with the different composite series Vs shore hardness, it would be much easier to visualize
  • 3.1.2 line 250 the authors presented so many equipments in such details and now they say "a testing machine". Which one?
  • 3.1.3 I do not see the point of the presented FTIR. You see the gelatin spectra when you added gelatin and PVA when you added it, then? The authors get information about the already known composition, but do not discuss of the materials interactions in the composite or any other point.
  • 3.1.4 figure 4 is not clear. Line 305 what the authors mean with "best content of gelatin"?
  • 3.2.1 again just a list of results, no discussion at all is provided
  • 3.2.2 results discussion is presented, more references from the literature are necessary as well as an english check.
  • 3.2.3 I do not see the TGA data. I would like to see the results of the composite without fillers to make a comparison of their effect. How many cycles were performed with the DSC to get the presented curves?
  • line 387 is not clear to me what the authors mean
  • 3.2.5 rather then crosslinking, it is not the hygrosopic nature of the materials that has the stronger effect?

Line 35 extra characters in "bottled water"

line 82 In comparison: the I is missing

Author Response

Responses for the Reviewer 1

We would like to thank you for all the comments, we have introduced corrections in the attached text of the article, as well as in the answers below.

  1. Both in the summary and in the introduction, the terminology was changed and attention was paid to assigning the term biodegradable to the right raw materials and additives. Changes are marked in yellow.
  2. Research methods section has been changed. Composite preparation, mechanical characteristics and surface analysis were modified - removing redundant information.
  3. The analysis of the results section has been changed. A diagram has been introduced showing the mechanism of the probable interactions between starch and gelatin. Literature has been added.
  4. In chapter 3.1.1. hardness chart has been added.
  5. In section 3.1.2 the lack of data on the testing machine has been corrected, manufacturer, country of origin added.
  6. The introduction of the fTIR spectrum of gelatin and gelatin and PVA has been corrected and discussed, the mechanism of probable interactions between these biopolymers has been shifted, and the different properties of PVA in the context of goniometric analysis and greater wettability of the surface against polar liquids compared to other composites have been discussed.
  7. Indeed, this is an imprecise phrase about the "best gelatin content" (3.1.4). Has been changed in the text of the article.
  8. The discussion has been added.
  9. The sections related to analyzes in FTIR have been changed, reports from the literature have been introduced.
  10. DSC cycle count is 6, no TGA tests were performed.
  11. The fragment has been modified
  12. Indeed, the ability to absorb moisture and bind water from the environment for these composites is of paramount importance, therefore the hygroscopic effect of the materials was taken into account.

Line 35 has been changed to "bottled water"

Line 82 with the lack of "I" was supplemented: “In comparison"

Reviewer 2 Report

Manuscript title: Biopolymer composites as an alternative to materials for the production of ecological packaging

The manuscript submitted by author related to synthesis of the porous gelatin composites based on the economical solvent casting method in order increase the biocompatibility of the matrices with additives, where no organic solvent is used was the good effort made by authors. But still manuscript lack in several ways. Few of the concerns and suggestion are listed below.

  1. The yield and purity of all compounds must be reported, including the methods used to determine them. 
  2. Check the sentence of line number 82 “n compar- ison, materials containing synthetic compounds, so-called plastics, decompose over hun”
  3. Methods of purification used to prepare samples for characterization should be described.
  4. The use of self-citations is not permitted. Kindly check the refrences 26 and 27. 26. ProchoÅ„ M., Zaborski M., Masek M., Sposób wytwarzania kompozycji polimerowej z przeznaczeniem na opakowanie oraz 594 sposób wytwarzania kompozytu przeznaczonego na opakowania z kompozycji otrzymanej tym sposobem, P.429957, WIPO ST 595 10/C PL429957, 3175w, 2019-05-17 596 27. ProchoÅ„ M., Dzeikala O.: Sposób wytwarzania żelu polimerowego o ograniczonej palnoÅ›ci oraz sposób wytwarzania 597 kompozytu żelowego P.434901, (05.08.2020r.). Even other carefully.
  5. Provide the molecular weight of gelatin, potatoes starch etc used in this study. Evidence of molecular weight should be provided.
  6. Section 2.2. Preparation of the Composites, 2.4. Preparation of compositions based on gelatin, If Methods already published should be indicated by a reference: only relevant modifications should be described.
  7. Provide the strong reason for adding Casein and red phosphorus as fillers? With proper justification and how the ratio and concentration of each filler will further influence the composite.
  8. Figure 2. Tensile strength (a) and elongation at break (b), provide with error bar.
  9. Provide the schematic mechanism for the synthesis of composite in this script.
  10. Provide the structure composition, linkages of gelatin and potatoes starch.
  11. Conclusion section should be rewritten again.  The Conclusion should not be a summary, but should illustrate the advances and claims of innovative aspects of the research work done.

Author Response

Responses for the Reviewer 2

We would like to thank you for all the comments, we have introduced corrections in the attached text of the article, as well as in the answers below.

Description of composites processing methods along with the solvent casting method has been added in the Introduction section.

1. The purity of all reagents is stated, all reagents are polluted from the manufacturers indicated, and each is provided with a safety data sheet.

2. The sentence of line 82 has been changed.

3. The purity of all ingredients used has been added along with additional information included in the safety data sheets from manufacturers.

4. Citations of patent applications and patents have been removed. The fragment has been changed.

5. Molecular weightof the substances used were added in accordance with the manufacturer's data sheets.

6. Sections from 2.2. to 2.4 regarding the preparation of composites have been changed.

7. Casein, like red phosphorus, contains phosphorus atoms in its structure. . Due to their structure, both additives are included in the group of flame retardant compounds for polymeric materials, in addition to such as melamine, melamine phosphates, poly (melamine phosphates), borates, isocyanurates, poly (ammonium phosphates), etc.

8. Error bars in Chart 2 have been added.

9. The composite synthesis was introduced in the article (2.3. Selection of Matrix Composition).

10. The structure of probable interactions in macromolecules between gelatin and starch is attached to the article in section FTIR analysis.

11. I agree with the Reviewer's comment, the application section has been rewritten.

Reviewer 3 Report

In the manuscript entitled “Biopolymer composites as an alternative to materials for the production of ecological packaging” the authors have created a gelatin-glycerin and PVA-glycerin film composites with highest resistance to flammability, increased thermal stability, flexibility, greater hardness compared to other composites.  The work is of high quality and the manuscript is very well written and easy to follow. The dataset the authors have generated should be valuable for the production of ecological packaging field.

It is clear that the gelatin-glycerin films were found to exhibit higher cross-linking density compared to the films containing PVA and were more stable dispersions. However, what is the explanation for the inferior performance of each virgin materials compared to blends? All three components are difunctional and similar reactivity would be expected. The difference in performance should be discussed.

Author Response

Responses for the Reviewer 3

We would like to thank you for all the comments, we have introduced corrections in the attached text of the article, as well as in the answers below.

The conclusions were rewritten once again and the imprecise sentence regarding the cross-link density was removed. An additional scheme of interaction in the gelatin-starch structure was added, the research results were analyzed, the methodology was changed, etc.

Round 2

Reviewer 1 Report

I thank th authors for their answers and work on their manuscript. I find it now suitable for publication after minor revision. I suggest a general language review and some specific comments hereafter reported.

line 14 "biodegradable" is used twice.

line 117 what the authors mean with "Casting can be gravity..."?

line 130 I would use "pre-environmental" rather than "pro-ecological".

line 556 what th authors mean with "elegant structure"?

Some reference have a DOI, others do not. The authors hould armonize the bibliography following the journal guidelines.

line 691, 692 could the authors clarify the funding?

691 alization, O.D.; supervision, M.P.; project administration, O.D.; funding acquisition, M.P.

692 Funding: This research received no external funding.

Author Response

I would like to thank you for the review and send me some comments from the second round:

  1. (line 14) IIndeed of "biodegradable" it should be bio-decomposable".

2 (line 117) Zdanie dotyczy odlewania rozpuszczalnikowego które może takze odbywać siÄ™ metodÄ… grawitacvyjnÄ… lub rotacyjnÄ… przy uzyciu maszyn odlewniczych. Zdanie zostaÅ‚o przeorganizowane.

The sentence is about solvent casting which can also be done by gravity or rotation with casting machines. The sentence was reorganized.

  1. Line 130 Pro-ecological” has been changed”
  2. Line 556 The sentence has been changed
  3. The bibliography has been revised in line with the journal requirements
  4. The research was carried out within the statutory funds of the Lodz University of Technology

7.Line 691 Regarding visualization and etc. has been corrected

8.Line 692 This study did not receive external financing, e.g. project financing, but the statutory funds of the Lodz University of Technology were used.

Reviewer 2 Report

The authors have satisfactorily revised the script.

Author Response

Thank you very much.. I am attaching the incremental yellow amendments that have been corrected
